# Nature is resource, playground, and gift: What artificial intelligence reveals about human–Nature relationships

**Rachelle K. Gould** [1,2]*, **Bradford Demarest**[2,3], **Adrian Ivakhiv**[1,2], **Nicholas Cheney**[2,3]

**1** Rubenstein School of Environment and Natural Resources, University of Vermont, Burlington, VT, United States of America, **2** Gund Institute for the Environment, University of Vermont, Burlington, VT, United States of America, **3** Department of Computer Science, University of Vermont, Burlington, VT, United States of America

* rgould@uvm.edu

## Abstract

This paper demonstrates how artificial-intelligence language analysis can inform understanding of human–nature relationships and other social phenomena. We demonstrate three techniques by investigating relationships within the popular word2vec word embedding, which is trained on a sample from over 50,000 worldwide news sources. Our first technique investigates what theory-generated analogies are most similar to nature:people. The resource:user analogy is most similar, followed by the playground:child and gift:receiver analogies. Our second technique explores whether nature-related words are affiliated with words that denote race, class, or gender. Nature words tend slightly toward associations with femininity and wealth. Our third technique demonstrates how the relationship between nature and wellbeing compares to other concepts' relationships to wellbeing—e.g., spirituality–wellbeing, social relations–wellbeing. Nature is more semantically connected to wellbeing than money, social relations, and multiple other wellbeing correlates. Findings are consistent with previous social science and humanities research on human-nature relationships, but do not duplicate them exactly; our results thus offer insight into dominant trends and prevalence of associations. Our analysis also offers a model for using word embeddings to investigate a wide variety of topics.

## Introduction

Our global society confronts a complex suite of environmental issues (e.g., climate change, land degradation and declines in ecosystem services, species extinctions), and our ability to effectively address them will largely determine our collective future. In this high-stakes context, better understanding human relationships with ecological systems, relationships that are at the core of environmental issues, is a crucial area of inquiry. As our society confronts these environmental issues, we are also experiencing rapid development of computational techniques and artificial intelligence (AI) that allow us to better understand and organize the world, in general and within social science in particular [1, 2]. In this paper, we leverage

**Data Availability Statement:** The data used in this analysis (the word2vec model of the Google News corpus) are publicly available at https://github.com/RaRe-Technologies/gensim-data. Code used in analysis is described in detail in the supplementary

materials and is publicly available at https://github.com/brad-dem/w2v_gnews_nature.

**Funding:** This study was funded by the Gund Institute at the University of Vermont (https://www.uvm.edu/gund). Authors RG, NC, and AI received the funding. The funder played no role in study design, data collection or analysis, decision to publish, or manuscript preparation.

**Competing interests:** The authors have declared that no competing interests exist.

burgeoning artificial intelligence techniques to investigate human–nature relationships. We demonstrate the power of AI approaches to reveal *implicit, collective* understandings—that is, understandings that are subtle (i.e., not explicitly expressed) and shared [3, 4]. We apply these approaches to human–nature relationships but suggest that they may be relevant to other topics as well [3].

In this paper, we address three enduring topics in the environmental field, but from the new perspective offered by natural language processing (NLP), a form of artificial intelligence. We first introduce the topics and their scholarly contexts, including associated questions and gaps in understanding. We then present results for each question. We conclude by reflecting on how NLP results compare to existing interdisciplinary scholarly understandings.

## Three enduring topics in environmental research

This study addresses three perennial topics related to "the environment" (Table 1). The descriptions below summarize a well-established literature that served as the basis for our exploration of novel NLP analyses.

**Human–nature relationships (what is the character of human relationships with the nonhuman world?).** Study of human–nature relationships (HNR) is an active area of research that draws on a variety of methods and epistemological perspectives; it includes approaches from philosophy to sociology to empirical psychology [5]. Underlying this area of study is the important recognition that "nature" is a complex and contested concept deeply rooted in European and North American thinking and histories [6–8]. For many communities around the world, the concept of nature does not exist because there is no obvious separation between humans and the rest of the world [9, 10], and scholars increasingly acknowledge this [11, 12]. Despite full awareness of this foundation, the term is a useful shorthand and is widely used in academic research. Human–nature relationships take many forms and look very different across cultures, and extensive literature documents this diversity. A great body of foundational anthropological and ethnobotanical work, for instance, provides deep descriptions of particular societies' understandings of relationships with nature [13–16]. Scholarship on religion demonstrates that different belief systems embody and espouse diverse conceptions of human–nature relationships [17–21]. Philosophical scholarship has emphasized the types of values relevant to HNR–for many years, the dyad of intrinsic and instrumental values, with the recent addition of relational values [22–24]. Psychological research, almost entirely (until recent years) conducted with Americans and Northern Europeans [25], has used the connectedness-to-nature and inclusion-of-nature-in-self scales to acquire self-reports of that concept [26–28] and Implicit Association Tests to understand that connection in implicit terms [29–31].

**Table 1. Summary of our three analyses.**

| Environmental Topic | General environment-related question | Our NLP question |
|---|---|---|
| Human–nature relationships (HNR) | What is the character of human relationships with the nonhuman world? | How well do research-generated analogies between nature and people (i.e., those found in the scholarly literature) match the nature:people relationship in the text corpus? |
| Racism, classism, sexism, and environmental thought and action | How does environmental thought and action interact with race, class, and sex? | Are "nature" words gendered, raced, or classed? |
| Contributors to human wellbeing | What role(s) does the environment play in human wellbeing? | How does the connection between nature and wellbeing compare to other constructs we know are connected to wellbeing? |

Each analysis is associated with an enduring topic related to understanding thinking and research about "the environment;" a general question in the environmental research field; and our specific NLP research question.

Recent work, much of it interdisciplinary, offers summaries and frameworks to make sense of the great diversity that empirical and conceptual work in so many disciplines has demonstrated and detailed. Muradian and Pascual [32], for example, offer a typology of seven human–nature relationship types: detachment, domination, devotion, stewardship, wardship, ritualized exchange, and utilization. Flint et al. [33] review empirical studies that create typologies of human–nature relationships and distill three important dimensions on which categorizations differ: positions of humans and nature with respect to each other; character of the human–nature bond; and perspectives on understanding of nature. Ives et al. [34] review extensive literature on human–nature relationships and suggest three overarching categories for human–nature connection: human–nature connection as mind, as experience, and as place. The work cited here, and much more, offers a huge array of possibilities of human–nature relationships (even as some contest the universalizability of both "human" and "nature") [35, 36]. We set out to determine which relationships are dominant in a text corpus of billions of words on the English-language Internet.

**Racism, classism, sexism, and environmental thought and action (how does environmental action interact with race, class, and sex?).**   Conservation and environmental action have notorious histories of classism, racism, and sexism [37–40]. Environmentalism, as associated with the construct of "the environment" that emerged in North America in the mid-1900s, has since its inception been associated in popular conceptions with white, upper-middle-class, highly educated people [41]. This association is clear in perceptions of the average environmentalist as white, middle-to-upper-class, and male [42, 43]. It is also borne out in the employment rosters of conservation organizations, which are overwhelmingly white and until recently were highly skewed toward male, especially in leadership positions [37]. Though in recent years the gender gap has narrowed (women are now well represented in general, though not in higher-ranking positions), racism and classism remain persistent problems [44]. These relationships arguably run very deep—from game reserves for nobility in early modern Europe to the white-masculinity-infused wilderness narratives of the 1800s American West [45].

Yet another, largely separate, body of work suggests that nature connotes different associations—not with dominant demographics (white, male, wealthy), but with non-dominant ones (non-white, female, poor). In this second complex body of work, nature is often associated with femininity (e.g., mother nature) [46], Indigeneity [47, 48], and blackness [49], and it is relatedly understood as inferior to masculinity, civilization, and whiteness [50]. It is not clear which of these sets of associations, which largely pull in opposite directions, is most dominant in current popular discourse; in our analysis, we explore whether the environment is, in popular discourse, more associated with one side of a series of simplified dichotomies.

**Contributors to human-wellbeing (what role does the environment play?).**   Scholars have addressed questions of the good life, and what makes for a good life, for millennia. Approaches to characterizing the constituents of wellbeing are varied, but a few have received extensive attention, especially as related to global (human) development. The capabilities approach, which has received extensive international attention [51], suggests that wellbeing requires opportunities for core capabilities such as emotions, affiliation, security, and living in relation to other species [52, 53]. The Human Needs approach advocates for a human-scale development and identifies eight core human needs including identity, affection, and participation [54]. Environmental concerns were typically not central to these discussions, in that the interactions between these constituents and the environment were not heavily considered (e.g., "other species" was added to the capabilities approach after Sen's initial proposal of it). But starting around the new millennium, with the rise of the ecosystem services concept, scholars began to connect ecosystem function and human wellbeing in earnest [55]. Ecosystem services are ecosystem functions that benefit human wellbeing; the core idea behind the concept

is that human wellbeing is inherently and unavoidably dependent on ecosystem functioning, for both physical reasons (e.g., food, fiber, shelter) and intangible reasons (e.g., spiritual fulfillment, identity, heritage) [56, 57]. Though environmental scholarship clearly demonstrates close connections between nature and human wellbeing, whether these connections are widely understood, and thus are revealed in large bodies of text from mainstream sources, is unclear. We explore how widespread understandings of these connections may be.

## Natural language processing can provide unique insight into collective perceptions

Human perceptions, especially collective perceptions, are encoded into our language [3, 58, 59]; language is thus a powerful source of insight into those collective perceptions. Word Embeddings, a form of artificial intelligence for Natural Language Processing, are now sophisticated enough to allow us to analyze gigantic amounts of text to explain the variability and meaning encoded into our language. When algorithms are being used to design actions and influence activities in the world, this is a giant source of bias and substantial problem [60]; it has inspired important efforts to recognize and then mathematically reduce these biases [61]. However, this bias detection/embodiment is also a tool to increase understanding about collective concepts and how we as humans, at least humans who produce and use particular bodies of text, see and organize the world [3, 4].

Building on this idea of bias detection as a tool for deeper understanding, social scientists increasingly use word embeddings to add insight to our understanding of many nuanced issues. The present study is, to our knowledge, the first that applies word embeddings to questions of human-environment relationships. The process of creating word embeddings places each word in a corpus at a specific point within a metric space. This allows researchers to examine the distance, or other geometric relationships, between individual terms (or groups of terms, in the case of analogies). These geometric relationships, in turn, provide insight into the collective understandings embedded within human writing. Examples include finding that the concept of person/people is much more related to men than women [4] and that personal characteristics of warmth and competence are highly correlated, despite often being opposed in explicit reports [62]. The latter analysis, of warmth and competence, provides evidence that word embeddings represent implicit, rather than explicit, associations between words. This is a crucial point for analyses that use word embeddings to understand social phenomena, and it suggests one of the particularly powerful aspects of the technique: that it demonstrates associations that are both implicit and collective; social science has no other automated technique to understand this type of association.

In addition to revealing important insight about implicit word associations, temporal NLP analyses–i.e., those that analyze relationships over time–demonstrate that the algorithms reflect social trends. Examples of findings from temporal analysis of word embeddings include detection of changes in occupations as more associated with men vs. women or with different ethnic groups (the latter aligning with immigration patterns) [63], and that associations between the various dimensions of class have remained largely consistent in the 20th century, except for education which has increased in its independent association with affluence [3].

Here we develop and apply NLP tools to approach social science questions in novel ways. We build on existing uses and analyses of NLP models [63] to develop three techniques to elucidate implicit collective understandings evident in our text corpora. First, we explore how similar our target relationship (nature:people) is to a suite of candidate analogies drawn from the academic scholarship on human–nature relationships. Second, we analyze where our target words (nature-associated words) fall on a series of socio-demographic spectra (race, gender,

wealth). Third, we assess whether our target concepts (nature and wellbeing) are more or less connected than other concepts known to be connected to our inquiry concept.

Importantly, though we demonstrate how these techniques can provide insight into human–nature relationships, we suggest that they will be highly relevant to many other questions. We thus demonstrate how humans understand their relationships with nature according to billions of words on the Internet, but also provide examples of how natural language processing techniques can be used to address many issues, especially those beyond race and gender stereotypes that have dominated much social science use of AI.

## Materials and methods

### Overall design

We use multiple natural-language-processing techniques (artificial-intelligence-aided text analysis techniques) to understand the responses to our questions (Table 1) in a large-scale dataset. We focus on word embedding techniques, a common type of natural language processing. Word embedding techniques center around the ability to demonstrate, with great nuance, when words are similar in meaning. The most important aspect of word embeddings is that they attempt to capture meaningful semantic relationships from word contexts—the idea that, as Firth [64] said, "you know a word by the company it keeps." Much recent social science work that employs artificial intelligence (including nearly all of the work cited above) employs natural language processing, and in particular word embeddings. This is likely due to both the richness encoded in word embedding vectors–i.e., their ability to capture such nuance in words meaning and uses–and the rapid development and refinement of word embedding techniques.

Word embedding analyses depend upon the unique vectors that AI creates for every word in a text dataset. These vectors contain extensive information about the inferred semantic meaning of each word, as defined by the corpus on which the AI is trained. One of the most useful forms of information the vectors contain relates to each word's relationships with other words, as noted above; the AI attempts to understand the meaning of words by predicting omitted words from their context during training. Words that are used in more parallel contexts have more closely related vectors, indicating some form of similarity. We focus solely on vector similarity in our third analysis. As methods have developed, additional possibilities have emerged, and our analyses capitalize on these abilities. In particular, we have gained abilities to: indicate analogies that are more or less similar to one another (employed in our first analysis) and isolate particular elements of similarity (employed in our second analysis).

### Data source

The word2vec model used in this paper's analyses was trained on a corpus of roughly 100 billion words taken from the Google News aggregator site. This training yielded a model that includes three million unique words and phrases, each represented by a vector in 300-dimensional semantic space (https://code.google.com/archive/p/word2vec/). Though the training corpus (i.e., the text upon which the model was trained) is not available for public access, some information about the contents can be construed. Google News is a site that aggregates online content that Google classifies as news. In addition to media sites (e.g., CNN, the New York Times, WIRED), Google News also provides press release content from sites operated by governmental units (e.g., NASA), public and private companies (e.g., Facebook, Nintendo), and other organizations (e.g., Cato Institute, Brookings Institution). Information is collected and ranked algorithmically from news sources across the world, based on criteria including volume of site production, length of articles, usage patterns, number of news bureaus, and human

opinion of news sources, among others [65]. This results in a corpus skewed toward European and North American news sources [66]. Texts included most likely reflect written, rather than spoken, language conventions, although transcribed interviews could also be included if they appeared in a format classified as news. In terms of tone, the contents of Google News tend toward the formality of professional, edited reporting. Information about timeline is not provided, but it is likely from the advent of the internet until 2013.

## Word clouds for complex concepts

One notable element of our approach is our use of groups of words to indicate focal constructs. In NLP, thousands of factors influence each word's vector; one important factor is that many words have multiple meanings (e.g., black is a color and also a term to identify race). Because of this complexity, NLP data are notoriously complex–vectors for a given word are often imprecise when taken in isolation. For this reason, we follow a recently established convention in research that employs NLP to understand social patterns: we use "word clouds" (i.e., groups of words). These word clouds collect a small set of words closely related to the construct at hand. In aggregate (i.e., when their vectors are averaged), they provide a more accurate sense of the construct than any single word, with all the noise associated with it, would [67]. We created word clouds for each construct used in analyses 2 and 3 (e.g., we added words such as "african" to "black"). See Fig 1 and the S1 File for details.

## Statistical analysis

We conducted three analyses: an analogies analysis; a spectrum analysis; and a conceptual similarity analysis. Here we briefly summarize each analysis; the S1 File include extensive detail. Notably, for each of our three analyses, we provide "reference terms" that are unrelated to nature, to provide a sense of the similarity scores for "average" or random words or analogies and thus allow a comparison with the relative strength of the similarities we find in our analyses.

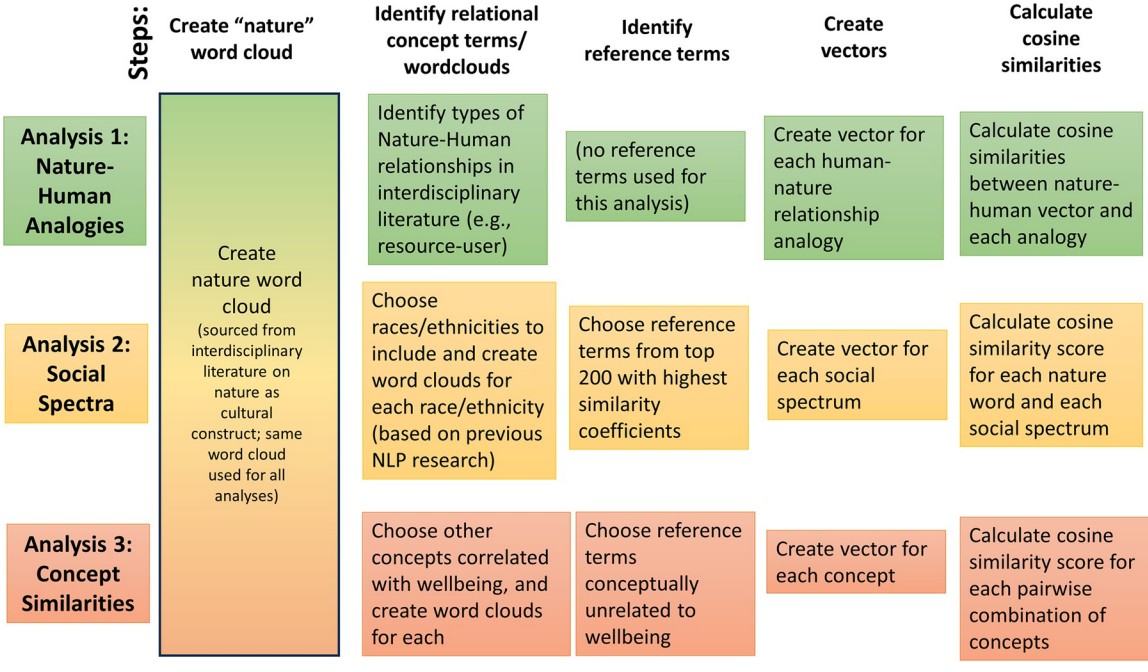

**Fig 1. Visual representation of methods.**

For our analogies analysis, we used existing techniques that characterize the similarity of two-word relationships. These analogies were some of the first and most impressive examples of the power of word embeddings (e.g., that the AI could determine that king:queen had the same relationship as man:woman) [68]. For this analysis, we created a list of human–nature relationships analogies based on our combined social science and humanities knowledge, and via analysis of existing typologies of human–nature relationships, as described above. We then compared these literature-derived analogies to nature:people and nature:humans. Specifically, we calculated 28 analogy scores for each human–nature relationship analogy: we calculated the similarities between all of the words in our "nature" word cloud and "people" and all of our nature-world-cloud words and "humans."

For our spectrum analysis, we built upon and modified a previous technique designed to elucidate the biases inherent in word embeddings. This previous work [61] created a male-female spectrum and showed, for example, that "lipstick" scored near the female pole and "tactical" scored near the male pole. We wished to see how nature-related words would place on that spectrum, and on other spectra of social groupings. We thus used a different previous work [67] to create small word clouds for each pole of multiple social groupings concepts: gender, wealth, and three ethnicities (black/white, Asian/white, and Latinx/white). We then calculated the spectrum score for each of our nature-word-cloud words on each spectrum, along with the average score for the nature words.

For our conceptual similarity analysis, we use a technique employed by many previous word embedding studies: we calculate cosine similarities between pairs of concepts [4, 63]. We choose concepts commonly identified as connected to wellbeing, and compare their semantic similarity to wellbeing to the semantic similarity between nature and wellbeing.

## Results

Our three analyses provide different insight on collective, implicit understandings of human-nature relationships. Below we detail results of those three natural-language processing analyses.

### The nature:people relationship

In this analysis, we address the question: ***How well do theory-generated analogies between nature and people match the nature:people relationship?***

We tested twelve analogies drawn from the interdisciplinary literature on human–nature relationships. The resource:user analogy most closely matches the nature:people analogy. The second highest match is playground:child, closely followed by gift:receiver (see Fig 1).

### Is nature raced, gendered, or classed?

In this analysis, we address the question: ***Are "nature" words gendered, raced, or classed?*** [i.e., if we create a "spectrum" of similarity between two aspects of gender, race, or socio-economic status, do "nature" words fall closer to one or the other?]

This analysis demonstrates that on balance, our nature words are more connected with "female" than with "male" and more connected with "rich" than "poor". They are not highly connected with any of the ethnic categories we explored. There are two ways to interpret the strength of the average score of the "nature" words. One is to compare the nature words' scores, and particularly their mean, with the reference words provided, which load strongly on each spectrum (see Fig 2). A second is to observe where the nature words' mean score falls relative to the overall distribution of the 50,000 most common words on each spectrum (the blue line on each diagram). Here, we find that the mean nature term vector is: more white (when

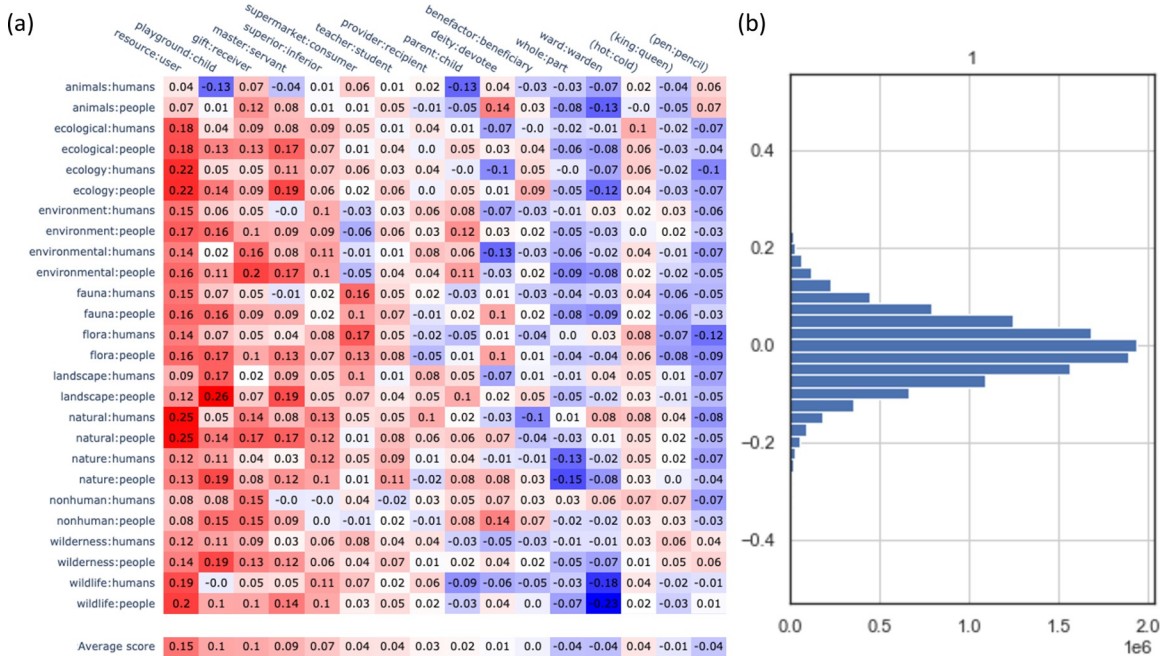

**Fig 2. Results of analogies analysis.** (A) Heatmap of similarity of literature-generated analogies to many possible permutations of the nature:people analogy (using all of our nature terms, each paired with both "people" and "humans"). The sign of each similarity index indicates the direction of the analogy: if the number is positive, the analogy matches more closely in the direction listed above; if the number is negative, the analogy matches more closely in the reverse direction. (B) Histogram of cosine distributions between the combined nature-people wordcloud concept vector and all combinations of the top 5000 terms in the corpus by frequency; this provides context for how strong relationships are between analogies (e.g., it demonstrates that cosine similarities of 0.2 are extremely uncommon).

compared to black) than 39.8% of the top 50,000 most common words (hereafter shortened to "words"), more white (when compared to Asian) than 35.5% of words, more white (when compared to Hispanic/Latino) than 59.9% of words, more female (when compared to male) than 79.6% of words, and more rich (when compared to poor) than 81.0% of words. Said another way, the mean nature term vector falls at roughly the 40th percentile of Black vs. White, with 0 being extremely closely aligned with "Black", the 50th percentile being exactly equally aligned with Black and White, and the 100th percentile extremely closely aligned with White. Thus the nature mean is slightly more aligned with the concept of Black as race/ethnicity than with White as race/ethnicity, but falls near the middle of this ranking with respect to the most common 50,000 words in the corpus. Roughly the same is true for the spectra that compare White to Asian and to Hispanic/Latino–though Asian is the furthest toward non-White. We see stronger results for the gender and wealth spectra. The mean nature term vector falls at roughly the 80th percentile of Male vs. Female (with 0 aligned with "Male" and 100 aligned with Female). Thus the nature mean is substantially more aligned with the concept of Female than with the concept of Male–more so than roughly 80% of the most common words in the corpus. The same is true for the wealth spectrum, except that the nature mean is more aligned with the rich end of the spectrum than 81% of the most common words (see Fig 3).

## Nature and wellbeing

In this analysis, we address the question: ***How does the connection between nature and wellbeing compare to other constructs related to wellbeing?*** We also provide four "reference" constructs to give a sense of similarity of less-related constructs.

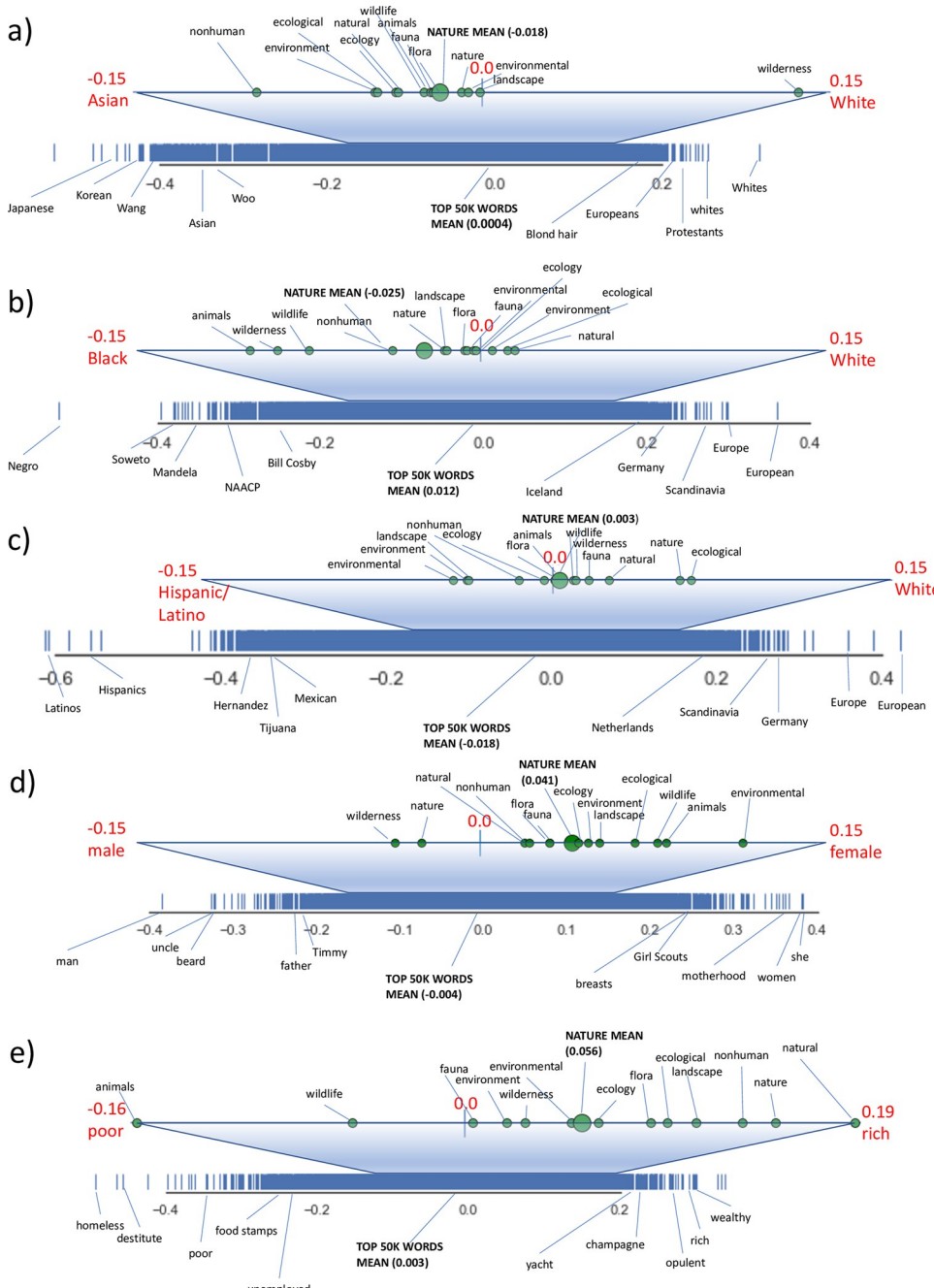

**Fig 3.** Depictions of the valence of nature words on the following scales: (A) Asian-white; (B) black-white; (C) Latinx-white; (D) male-female; (E) poor-rich. The trapezoid atop each scale represents a magnification of the part of the scale from -0.15 to 0.15. The summary statistic for each analysis (the mean of the values for each nature word) is labeled as "NATURE MEAN". The rugplot below each spectrum provides multiple reference points that help to interpret the nature-related results. It demonstrates the distribution of the 50,000 most common words in the corpus along each spectrum (i.e., it provides, for each spectrum, the average and range of values for the top 50,000 most common words). It also includes "reference words," in the form of **(A)** the single word that loaded most strongly on each end of the spectrum (e.g., Japanese and Whites for Asian-White), and **(B)** four additional words on each side of the spectrum, chosen from the top-scoring 100 words on each spectrum pole as particularly informative.

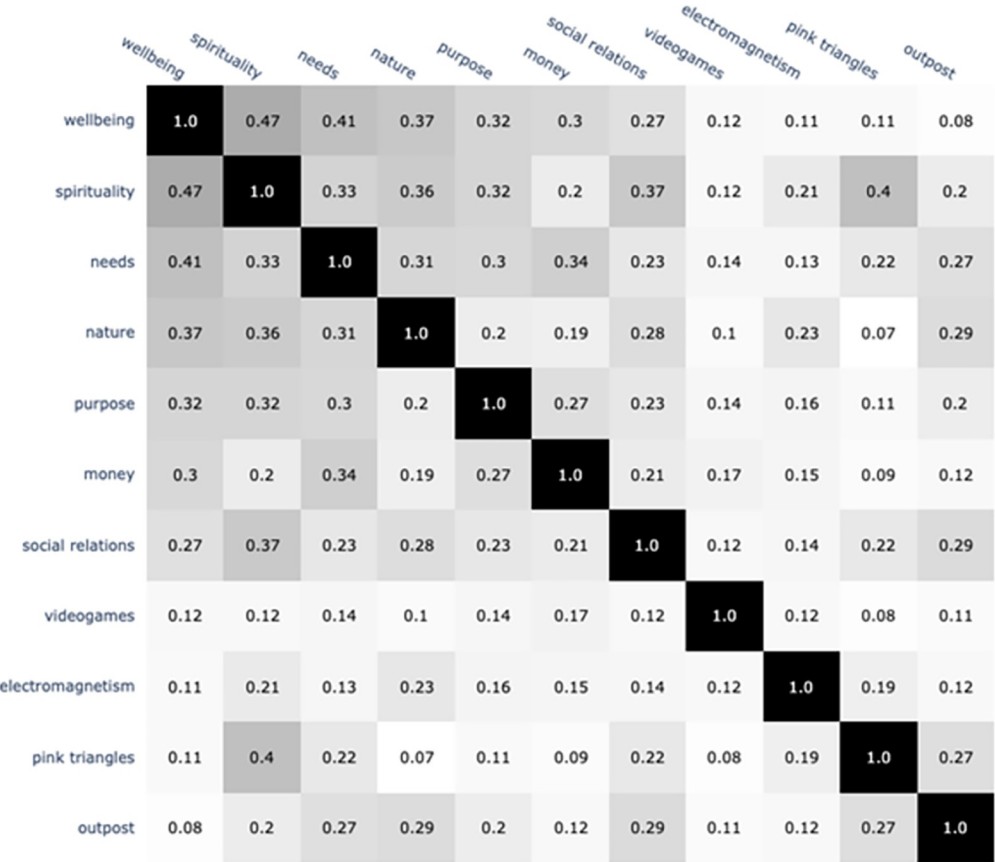

**Fig 4. Cosine similarities between the word clouds for wellbeing and a suite of potential correlates of wellbeing, as suggested by the literature.** Four terms have also been added for reference; we chose two (electromagnetism and videogames) because we anticipated they would have weak relationships with wellbeing; we choose two (pink triangles and outpost) randomly.

This analysis demonstrates how closely nature and wellbeing are associated, compared to many other constructs that the literature suggests would be associated with wellbeing. Of the potential contributors of wellbeing investigated, nature is the third-most similar to wellbeing. It outranks money, purpose, and social relations. The construct most closely associated with wellbeing is spirituality; this is closer to wellbeing than is nature, and by a substantial margin (see Fig 4).

## Discussion

Scholars study human–nature relationships from many angles; our approach uses artificial intelligence techniques (natural language processing) to reveal large-scale and deep (i.e., collective, implicit) patterns in how nature is referred to in an enormous sample of the English-language internet. Our results, which explore subtle, foundational issues related to human-environment relationships, dialogue with decades of in-depth scholarship in the environmental humanities in complex ways. Our AI-assisted results are consistent with broad trends and understandings in existing research, yet they add important new insight about relative prevalence and implicit collective understandings (in our dataset).

Our results inform ongoing, crucial societal conversations about the roles AI will play in society and in academic research in particular. In this study, we use AI to help us understand

subtle, deeply rooted phenomena typically studied largely by the humanities. As the use of AI in research was ramping up in the 1990s, one scholar wrote that unlike traditional humanities research, AI-aided research does not require a "virtuoso" human interpreter [69]. Our experience suggests an update to this claim, and this update aligns with emerging research in the digital humanities, which employs AI in various ways [70]. Our experience suggests that effective use of NLP in research still requires expert human input, at two steps in the process. First, human expertise is needed to determine what to study, based on existing knowledge (here, this was our selection, from the literature and based on our expertise, of human-nature relationships and contributors to wellbeing). Second, human expertise is needed to make sense of results–to explain them, including "ground-truthing" them via comparison to existing understanding. In other words, meaningful use of AI in research requires, we suggest, human expertise that can not only understand the immense complexity at play, but detail that complexity at every step of the interpretive process. The AI thus complements human expertise by making it possible to analyze much larger data sets, which can reveal enormous-scale implicit associations in new ways; human analysts can then explore whether and how an AI analysis corroborates the more nuanced, but selective, analyses made by human interpreters.

Below, we summarize how our AI results interface with existing understandings of human–nature relationships, the raced/gendered/classed nature of the environment, and wellbeing-nature links.

### Human–nature relationship descriptions/types

Our analogies analysis lends insight into implicit collective understandings of the human–nature relationship as revealed in our data source. Here we discuss how the three analogies that our analysis found to be most similar to nature:people relate to the academic literature.

The resource:user analogy, which dominates our results as an analogy closely connected to the nature:people relationship, aligns with perceptions of nature as natural resources. This framing of nature emphasizes that nature provides goods that humans use, and it is a highly instrumental way to conceptualize human–nature relationships. It is strongly associated with economic thinking [71, 72] and "carries a sense of 'free goods' [and] human-centric use and commodification of nature" [73]. In mainstream Anglophone literature and culture, this perspective has arguably been dominant for hundreds of years [72, 74]. In recent decades, however, environmental scholars have identified this term as an incomplete and potentially detrimental way to refer to nature [73].

Critiques of the "natural resources" mindset are many. Some scholars argue that the resource concept assumes instrumental usage and utilitarian valuation, thereby precluding intrinsic values, subjectivity or co-agency of nature, and relations of obligation, respect, aesthetics, or tradition [22, 23, 75]. Other scholars point out that "resourcism" does not deal effectively with finitude, intergenerational equity, or other "externalities" [76–78]. Scholars suggest, in other words, that this framing needs changing if we are to reach sustainability. That this analogy was by far the most strongly aligned with nature:people in our study is not surprising, but it confirms, without the bias a human analyst might introduce, a need for a chance in perceptual tendencies. It suggests that the general implicit ethos in the English-speaking, internet-using world is most aligned with the view that nature is a resource for human use–a view widely recognized as not particularly conducive to sustainability.

The playground:child analogy, which was substantially less similar to nature:people than resource:user but the next most similar analogy, aligns with perceptions of nature as a space of leisure and recreation. A leisure-based approach to nature is often associated with the same instrumental mindset the "natural resources" viewpoint entails: nature exists to serve human

needs. That the playground:child analogy was second-most similar to nature:people offers new insight into how deeply the concept of "nature as place of recreation" has infused English-speaking, internet-using culture.

Scholars occasionally refer to nature specifically as a playground for children [79, 80], but this analogy was designed to reflect the broader concept of nature as a space for recreation, leisure, and health promotion. Recreation and leisure are arguably the dominant ways in which upper- and middle-class Americans and Europeans interact with the natural world [81–84]. Historical precedents include the 18th-19th century European creation of game reserves and other privatized lands for hunting and other recreational uses by landowning nobility and colonial travelers [85, 86], and late 19th-early 20th century physical and "moral" health-promoting movements such as scouting, woodcraft, camping, and hiking [87, 88]. Recreational uses of nature have in turn become a strategy for rural development in formerly agricultural and post-industrial landscapes across North America and Europe [89–92]. Since the mid-20th-century, a combination of factors including increasing urbanization, racial desegregation and diversification, highway-building, and rising incomes led to the phenomenon of ex-urban "white flight" wherein middle-class, primarily white Americans moved from cities to suburbs and exurbs to live and recreate "closer to nature" [89]. With more recent gentrification of inner cities, there has been some reversal of this trend, but this has also been associated with urban greening and park revitalization projects.

The gift:receiver analogy, which was slightly less similar to nature:people than playground:child, can be seen as an alternative to the resource:user analogy, and its somewhat high similarity score merits attention. The idea of "nature's gifts" involves understanding the necessary benefits that nature provides to people as gifts (rather than, for example, as services [56, 93, 94]). Gift-related language often arises in Indigenous discourse [95, 96]. Consistent with the idea of a gift, this perspective is often infused with gratitude for the gifts offered and an inclination toward reciprocity [97–99].

Many recent science-policy processes suggest that the "nature as gift-giver" mindset is more conducive to sustainability than either the resource-based or leisure-based conceptions [100, 101]. The recent shift in the global community from using ecosystem-services-related terms to "nature's contributions to people" was largely an effort to better encompass the idea of "nature's gifts" [100, 101]. The relatively high score of gift:receiver in our results is thus promising for transitions toward sustainability. It suggests, perhaps somewhat surprisingly, that reactions of gratitude toward nature (i.e., for receipt of a gift) are relatively common in English-speaking, internet-using society. Efforts to foster this viewpoint would, research suggests, likely lead to more sustainable approaches to human-nature interaction.

The three analogies most similar to nature:people in this corpus offer interesting insight into implicit collective understandings of the human relationship with nature. The two most similar analogies (resource:user and playground:child) both align with a largely instrumental perspective on human–nature relations: nature provides goods to humans–physical goods (resources) and non-use goods (recreation). Decades of scholarship and recent United Nations scientific reports suggest that a purely instrumental approach to human–nature relationships is unlikely to be consistent with long-term sustainability, and that integrating other understandings of nature's value will be more conducive to sustainable pathways [102–106]. Perspectives other than instrumental include those focusing on nature's intrinsic values [103, 107, 108]; its relational values, that is, values associated with the reciprocal relationships people have with nature [22, 57, 109–111]; spiritual/religious and Indigenous perspectives (which overlap with relational perspectives) [19, 112, 113]; and those that otherwise seek to overcome the duality of humans and nature in theory and in practice [77, 105, 114, 115]. The third most similar analogy in our analysis (gift:receiver) aligns more closely with relational and religious/

Indigenous perspectives. Though it is not overwhelmingly similar to nature:people, its relatively high ranking offers hope for a future that integrates more reciprocal relationships between people and nature [116].

## The raced/gendered/classed character of nature

Our analysis of social-group spectra created a "similarity spectrum" for pairs of social-group words (e.g., rich/poor; man/woman), then determined whether our suite of 14 nature terms was more associated with either end of each spectrum. Results help to elucidate the multifaceted nature of how "nature" relates to social group concepts. Indeed, an important complexity of this analysis is that it in some ways invites more questions than it answers: in particular, it does not (like our other analyses) indicate the nature of the relationship between any two concepts. We further discuss this issue below.

Past work related to all three dimensions we explored (race, gender, class) suggests that nature is associated with both ends of the spectrum–in different ways. For instance, nature is feminized in conceptions like "mother nature" and its nurturing qualities, but is associated with masculinity in wilderness-related conceptions [45]. While these may appear contradictory, they are historically related in that masculinity's association with both the "conquest" and "enjoyment" of nature (e.g., in Romantic wilderness quests or mountaineering) has been accompanied by ideas of women and nature as closely connected—both are motherly, "virgin," restorative, and so on [117–120]. The same bivalent association exists for race and class. On one hand, "whiteness," especially of upper-class white males, has often been associated with "civilization" and "culture," which are contrasted to nature—and non-whites and lower classes have been associated with the nature end of that culture-nature spectrum. On the other hand, the idea of nature or wilderness as a site of leisure and recreation is closely associated with middle- and upper-class European and Euro-Americans [121, 122], and these groups have historically been and remain over-represented in environmental movements, discourse, and policy-making [37, 38, 42]. This leads to widespread characterization of the environmental movement as overwhelmingly white–created by and perpetuated for the interests of white people and the upper classes [123–125].

Our analysis could therefore have revealed an implicit tendency toward one or the other of these poles. We did find this tendency for the gender-related and wealth-related analyses: our nature words are more closely aligned to the female pole and to the rich pole. The gender result suggests that deep embedded associations such as "mother nature" are more aligned with implicit collective understandings of nature than are associations such as "manly wilderness." Overall, people (who use the English-language internet) see nature as more feminine than masculine. Similarly, the wealth result suggests that understandings of nature as elite space– e.g., the outdoors as a place for the wealthy–are stronger than understandings of nature as less civilized and thus connected to lower social classes.

Unlike the results for gender and wealth, all three analyses of ethnicity show little tendency of the nature words toward one or the other pole. The fact that word-embedding relationships with ethnic categories were relatively weak has at least two potential explanations. The first is that the two "valenced" conceptions exist for the nature concept (i.e., it is associated with both white people and people of color, in different ways); the near-zero rating is thus a result of pulls toward each pole "cancelling each other out." A chronologically or otherwise distinguished sample could provide more data for interpreting how words' positions on the spectra may change over time or between contexts. A second possibility is that the poles are simply not highly relevant to or associated with nature. Comparing nature words' placement on the spectra to the scores of "reference words" helps to interpret the likelihood of this second

explanation. The question is, how strong are the nature words' rankings in comparison to words that load very highly on a particular pole? We can take the Black-White spectrum as an example. The words most aligned with the "Black" end of the Black-White spectrum are Negro (-0.52) and Soweto (-0.40). The words most aligned with the "White" end of that spectrum are European (0.36) and Europeans (0.30). Given that some of the nature words have scores of close to -0.10, these terms' associations with blackness seems non-trivial (though deep interpretation of these numbers remains difficult).

We thus think it likely that the near-zero scores for the race/ethnicity spectra result from a "cancelling out" effect, largely because examination of the spectrum placement of individual nature-related words is generally consistent with existing literature. For instance, the word "wilderness," with its deeply problematic associations and white, wealthy biases, loads more on the white (as opposed to Latino) and wealthy (as opposed to poor) ends of the spectra. Yet "wilderness" is closer to black than white. This could be seen as confusing and inconsistent with explicit associations with wilderness (e.g., that black people are less likely to feel comfortable in and therefore recreate in "white spaces" doing "white activities" like backpacking [122, 126]). Perhaps results for the race spectra capture the confusing relationships between historically rooted conceptions and complex actualities. For instance, "blackness" has historically been conceptualized, in colonialist discourses, as closer to the "state of nature" than to "civilized" humanity [49, 127, 128]. Yet "blacks" have also been forced out of "naturalized" spaces such as colonial game reserves, national parks, and other protected areas. Adding yet more complexity, even though perceptions indicate that black people are less likely to engage with places like national parks, many black people have rich, meaningful, and/or complicated relationships and practices with such "natural" spaces [129]. Given the wealth of literature interpreting the various relations between race and conceptions and practices of "nature," we posit that further and more nuanced research will reveal interesting correspondences as well as changes over time.

Closer analysis of our results suggests that the relationships that our spectra capture are not all of the same type; this leads to complexities in interpretation and invites future analysis (e.g. with rapidly evolving AI techniques that may better attend to such complexity). Some associations, for example, are those of use or interest: between nature as an idea (as we have elaborated it) and the people who hold that idea. These people, our results may suggest, tend to be more rich and more "male" than average. Other associations are those of perceived identity; for instance, women and the wealthy being identified more closely with the idea of nature. Our analyses cannot parse these (or surely other) types of association, but doing so provides interesting fodder for future development of additional word embedding techniques.

## Nature as a, but not the, top contributor to wellbeing

Our analysis of the connections between "nature" and wellbeing analyzed the semantic similarity between the nature concept and a suite of other concepts widely acknowledged as strongly connected to wellbeing. The primary result is that "nature" is closely connected to wellbeing—more strongly than money, life purpose, and social relations. Yet it is also not the top contributor, of the constructs we tested.

The relationship between nature and wellbeing provides another example of the consideration mentioned above: that the character of the association between the two concepts is general; it aggregates, or synthesizes, across all possible facets of the complex relationships between the two terms. Our results that indicate nature's important connection to wellbeing align with the recent blossoming of work that recognizes associations between nature and wellbeing. These associations are complex and multi-faceted. Extensive research and international synthesis, over many decades and spanning scales from local to global, demonstrates the diversity

and necessity of nature's contributions to people (in some contexts called ecosystem services) [55, 57]. Central to this work is recognition of the intertwined material and nonmaterial ways that nature contributes to human wellbeing. The tens of thousands of studies that underlie these conclusions demonstrate diverse physical benefits (from food to storm attenuation to water purification) [130, 131], along with non-material benefits that include cultural and identity-related phenomena [132, 133] and benefits to mental health and cognitive function [134–136]. Our results reflect the combination of these relationships–because they are all represented in the richness of the wellbeing concept. The same is true for the other constructs in our analysis.

Our results align with past research that demonstrates thick and deep ties between nature and wellbeing. Yet our results also suggest that nature is not, in the collective consciousness, the top factor associated with wellbeing. Results which indicate that "nature" ranks below only "spirituality" and "basic needs" in semantic similarity to wellbeing offer important insight and suggestions for future research. First, the significance of "spirituality" should be seen in the context of a history within which this term has been taken as a general indicator of a meaningful and motivated life [137, 138]. That is, one of the most important connotations of the concept of spirituality is a deep type of wellbeing, whereas the term nature has a broader set of meanings and associations. Studies of the history of spirituality reflect the term's gradual shift from being associated with specific forms of religion (particularly Christian) to being used in increasingly secular, "post-religious," and cross-cultural contexts [139, 140].

Second, that "basic needs" are more connected to wellbeing than nature is also not surprising. The connection between basic needs and wellbeing is almost semantic: the concept of basic needs (the terms we included in that word cloud, such as basic_necessities, food_stuffs, and basic_needs; see S1 File for full list) essentially by definition denotes things that are required for health and wellbeing. Considering that nature is connected to a wide variety of phenomena other than wellbeing (whereas "basic needs" are connected to very few other concepts, and spirituality is connected to relatively few other concepts), the high similarity score between nature and wellbeing indicates a fairly strong connection.

It is also helpful to note that the "nature" wordcloud is also closely connected to both "spirituality" and "basic needs"–though less strongly than it is to "wellbeing". This suggests that these three terms, in particular, are all highly related in implicit collective consciousness. This is not surprising, as nature clearly provides for many of humans' basic needs (e.g., food, shelter) and many people associate nature with spirituality and connections to something greater than themselves [19, 20].

The relatively weak connections between money and wellbeing invite reflection. This result suggests that though money may be (at least superficially) *touted* as of utmost importance in mainstream English-speaking society, collective implicit understandings do not see it as such. The most studied relationship here is between income and happiness: repeated studies show that income is weakly correlated with reported wellbeing (i.e., happiness) [141], but the relationship is extremely complex [142]. Perhaps most notably, an "income satiation" effect seems to exist, wherein the relationship between income and reported wellbeing breaks down, or may even reverse, at higher income levels [143]. In sum, though money is clearly connected, in mainstream English-speaking society, to status and prestige, our results suggest that it is less connected to wellbeing than more nuanced concepts of spirituality, nature, and basic needs.

### Novel uses of NLP techniques

We use word embeddings in slightly new combinations and ways, and thus add to the already wide variety of applications [144]. In this section, we describe this novelty. We suggest that these techniques, likely with slight permutations, could be applied to work on diverse topics.

Our analogies analysis adds a new dimension to the use of analogies in word embeddings. In past NLP work, analogies have been used to demonstrate the power of NLP, and as a benchmark to measure the ability of various word embedding techniques to capture known embeddings [145–149]. They have been used on straightforward, unequivocal analogies, such as between a country and its capital city. These examples demonstrate the richness of the vectors that represent each word, but they do not provide new information. Analogies have not, to our knowledge, been used as a tool to reveal implicit collective understandings of less clear-cut topics–for instance, of highly culturally determined relationships. Our suggested technique–of proposing a suite of widely accepted analogies for the target relationship, and seeing which most closely match according to a particular corpus–could be used to explore an enormous variety of relationships, from race:class, to weather:mood, to food:health.

Our spectra analysis similarly takes existing NLP work a step beyond its current applications. It also offers a new possibility for work on social groupings. Our spectra analysis is similar to many recent analyses, but it combines and extends these recent analyses. Many recent analyses calculate cosine similarities and interpret them in various ways (as in refs [4, 63, 67] and our conceptual similarity analysis), but our spectrum analysis employs a more nuanced and specific use of cosine similarities. In sum, we combine the common cosine-similarity-comparison tactic with a less common spectrum-between-two-concepts tactic that was created to de-bias word embeddings [61]. The novel combination allows us to investigate a concept that is arguably more complex than others analyzed; we can explore both how that complex concept's elements and its mean relate to two sides of a social-grouping spectrum. We suggest that this spectrum approach could be used to explore where almost any concept, and especially any complex concept, falls on the spectra–either the spectra we use (for social groupings), or thousands of other possible spectra of interest. This is crucial to understand, as the use of word clouds of multiple terms to represent a topic is a common approach employed in the cosine similarity analyses noted above, but the specific terms deemed to be representative of these topics are typically chosen by the experimenter ad hoc (though many studies (e.g., [4]) run sensitivity analyses to determine the impact of word-cloud composition). Our spectrum approach, in contrast, allows us to simultaneously analyze both topics and the terms that compose them, which provides a mechanism to investigate the natural variation and experimenter bias within groups of words representing generic topics explored in social science NLP investigations.

Our conceptual similarity analysis offers the least novelty from an NLP perspective. It follows a similar pattern to the many papers cited above, which use relatively unvarnished cosine similarities and connect them to a variety of important social-science topics.

## Next steps

**Asking not only if concepts are related, but how.**   Many existing applications of word embeddings to social science focus on analyzing whether, or to what extent, two concepts are related. In many ways, our wellbeing and spectrum analyses exemplify this trend. These two analyses, like many past AI analyses in social science, do not explicitly address the qualitative details of those relationships. For instance, our wellbeing analysis does not reveal the ways in which wellbeing is related to spirituality, basic needs, money, nature, etc.; our spectrum analysis does not reveal how nature is more related to femininity or wealth than it is to masculinity or poverty. Those analyses reveal solely whether those concepts are more or less semantically similar. The question "how" demands a more nuanced, complex answer than does "if/to what degree"—more complex both in terms of structure (which dimension(s) reflect words' closeness?) and in terms of interpretation/function (via which concepts and contexts would we as

humans understand the words' relationships?). Our analogies analysis comes closer to addressing the "how", by taking a different tack. The analogies function more obviously encompasses a greater breadth of the meanings and relationships that the vectors encapsulate.

This leads to a fascinating and central question for word embedding analyses and what we can learn from them: How can we more fully understand and represent the multi-dimensionality of the vectors? Word embeddings scholars identified the obscureness of the semantics concept as a core challenge of the field early in its development. One scholar summarized this challenge as a lack of clarity in "which type of relationships between words word embeddings should reflect, because there are many different types of relations between words" [150]. In our data, this manifests as, for instance, the fact that the association between 'nature' and 'femininity' may be a different *kind* of association from that between 'nature' and 'masculinity.' Past research (as cited above) suggests that 'nature' and 'masculinity' concepts occur together more often in contexts related to use, mastery, strength, domination, etc., while 'nature' and 'femininity' concepts occur together more often in contexts discussing vulnerability, unpredictability, uncontrollability, weakness, threat, etc. These "type of relationship" questions present an active area of inquiry for word embeddings research.

**Additional text corpora.** We reveal implicit collective understandings in a general sample of English-language internet; the next step for this work would be to explore similar questions using other text corpora. Full understanding of the diversity (or lack of diversity) of responses to our questions will require moving beyond the populations represented in our study—both contemporary and across time. The AI is learning about, and teaching us about, associations in this particular dataset. This means that the text corpus is crucial to any results. There is increasing attention to the fact that the text corpora used to train algorithms are very far from neutral, and both what they are and how they are created is of utmost importance and dramatically impacts results [151]. For this reason, one important way forward is to apply these techniques and the questions about human–nature relationships to other corpora (e.g., religious texts; Indigenous texts; corpora in other languages), and to compare results from mainstream English-language sources to these alternate bodies of text. As one example of the potential in this arena, recent work explores word associations in distinct text corpora comprised of documents complied from distinct organizational contexts. This work finds that hiring women into senior leadership positions is associated with shifts in language associations—specifically, a reduction in gender stereotypes such as weak associations between women and leadership-related qualities like agency [152].

A frontier for much social science work that uses NLP is to create new repositories of text to understand these patterns in culturally varied contexts. We also emphasize that the training data used for AI analysis are crucial (as are data sources in any research project), and use of new text corpora will need to consider this important step. Given the amount of work required to create usable corpora, along with the possibilities for diverse corpora to help us understand the diversity of human experience, we call for a collective effort to collect and curate corpora that represent varied perspectives and communities. A few examples of publicly available repositories already exist; see, for example, clinical (medical) datasets [153] and a repository of over 1,700 text classification datasets [154] that range from compiled Yelp reviews to 100K descriptions of cultural items from the Italian Cultural aggregator. As we consider a dramatically wider scope of such publicly shared resources, crucial questions include who will create them, who will fund those efforts, and how decisions are made as to what to make available. This work will require a new wave of ethical consideration, for instance of text ownership and intellectual property as related to gigantic textual datasets [151], and a specification of the who, the how, and the what is beyond the scope of this paper. Given the potential of what we can learn, especially about heterogeneity in collective conceptions, moving forward for ethical sharing seems an important next step.

## Supporting information

**S1 File.**
(DOCX)

## Acknowledgments

We are grateful to Google, which compiled the text corpora used in this analysis.

## Author Contributions

**Conceptualization:** Rachelle K. Gould, Adrian Ivakhiv, Nicholas Cheney.

**Formal analysis:** Rachelle K. Gould, Bradford Demarest, Nicholas Cheney.

**Funding acquisition:** Rachelle K. Gould, Adrian Ivakhiv, Nicholas Cheney.

**Investigation:** Rachelle K. Gould, Bradford Demarest, Nicholas Cheney.

**Methodology:** Rachelle K. Gould, Bradford Demarest, Nicholas Cheney.

**Project administration:** Rachelle K. Gould.

**Visualization:** Rachelle K. Gould, Bradford Demarest, Nicholas Cheney.

**Writing – original draft:** Rachelle K. Gould, Adrian Ivakhiv.

**Writing – review & editing:** Rachelle K. Gould, Bradford Demarest, Adrian Ivakhiv, Nicholas Cheney.

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
