## [Decision Letter · Decision Letter 0]

11 Sep 2023

PONE-D-23-21507Nature is resource, playground, and gift: What artificial intelligence reveals about human–nature relationshipsPLOS ONE

Dear Dr. Gould,

Thank you for submitting your manuscript to PLOS ONE. After careful consideration, we suggest minor revisions before we accept the paper for final publication. Both reviewers made minor suggestions and comments. Please consider both reviewers' comments and make edits accordingly. Also, check the paper's citation styles in the final submission. PLOS uses the reference style outlined by the International Committee of Medical Journal Editors (ICMJE), also referred to as the “Vancouver.”

We look forward to receiving your revised manuscript.

Kind regards,

Zihao Zhang

Academic Editor

PLOS ONE

2. In your Methods section, please include additional information about your dataset and ensure that you have included a statement specifying whether the collection and analysis method complied with the terms and conditions for the source of the data.

“We are grateful to the Gund Institute for the Environment, which provided funding for this project.”

“This study was funded by the Gund Institute at the University of Vermont (https://www.uvm.edu/gund). Authors RG, NC, and AI received the funding. The funder played no role in study design, data collection or analysis, decision to publish, or manuscript preparation.”

Reviewers' comments:

Reviewer's Responses to Questions

**Comments to the Author**

1. Is the manuscript technically sound, and do the data support the conclusions?

Reviewer #1: Partly

Reviewer #2: Yes

2. Has the statistical analysis been performed appropriately and rigorously? 

Reviewer #1: Yes

Reviewer #2: Yes

3. Have the authors made all data underlying the findings in their manuscript fully available?

Reviewer #1: Yes

Reviewer #2: Yes

4. Is the manuscript presented in an intelligible fashion and written in standard English?

Reviewer #1: Yes

Reviewer #2: Yes

5. Review Comments to the Author

Reviewer #1: From my view, the authors' methods are sound and their analyses effective. The writing is generally intelligible with few typos or errors. The general arguments and novel approaches have potential for great contribution to the subject areas in question. I am, however, recommending minor (possibly major) revisions to the structure of the manuscript. The main reason for this is that I believe the team is trying to accomplish too much in just one paper, given its current framing. In my understanding, the authors are attempting to 1)establish the robustness of human-environment relationships literature 2) provide a novel approach to analyzing that robustness through the use of NLP methods, particularly word embedding, and 3) introduce novel approaches to NLP borne of their own interdisciplinary model. Both subject areas they are contributing to (Human-Envi. Relationships and NLP) are quite deep, and I think the their stated goals for the paper in either space are not fully met. They either lack specificity or relative robustness in the results as explained, and thus need further expounding to convince me otherwise, which will be difficult to do in just one paper. This undertaking might deserve multiple papers in its current framing. I suggest the authors either reduce scope or reframe more deeply onto one or two aspects (e.g. just the demonstration of the NLP methods' effectiveness or just the robustness of the literature that produces their analogies and how those are better grounded through NLP). See attached pdf with more comments. In short, I think the work itself is sound and should be eventually published, just with more comprehensive argumentation/presentation.

Reviewer #2: Dear Authors,

I've had the opportunity to review your manuscript on the patterns of human-nature relationships as interpreted from an extensive internet dataset. Firstly, I'd like to commend you on the clear structure of your paper; it greatly facilitates understanding and allows the reader to follow your arguments with ease.

Your novel use of NLP techniques in the realm of social science is particularly commendable. The three specific techniques you've applied have not only provided fresh insights into the topic but have also showcased the promising potential of integrating advanced AI methodologies into traditionally qualitative fields. This interdisciplinary approach is both innovative and inspiring.

While the methodology is detailed, a minor suggestion would be to consider incorporating a workflow diagram or flowchart. Such a visual representation would offer a clearer, concise overview of your AI-assisted approach, aiding readers in comprehending the technical processes involved more readily.

In conclusion, your work is a valuable contribution to both social science and the growing field of AI research. I look forward to its further refinement and eventual publication.

6. PLOS authors have the option to publish the peer review history of their article (what does this mean?). If published, this will include your full peer review and any attached files.

Reviewer #1: No

Reviewer #2: No

---

## [Author Response · Author response to Decision Letter 0]

30 Oct 2023

I have uploaded a separate file with a table that details our responses.

---

## [Decision Letter · Decision Letter 1]

2 Jan 2024

Nature is resource, playground, and gift: What artificial intelligence reveals about human–nature relationships

PONE-D-23-21507R1

Dear Dr. Gould,

We’re pleased to inform you that your manuscript has been judged scientifically suitable for publication and will be formally accepted for publication once it meets all outstanding technical requirements.

Kind regards,

Zihao Zhang

Academic Editor

PLOS ONE

Additional Editor Comments (optional):

Reviewers' comments:

Reviewer's Responses to Questions

**Comments to the Author**

1. If the authors have adequately addressed your comments raised in a previous round of review and you feel that this manuscript is now acceptable for publication, you may indicate that here to bypass the “Comments to the Author” section, enter your conflict of interest statement in the “Confidential to Editor” section, and submit your "Accept" recommendation.

Reviewer #3: All comments have been addressed

2. Is the manuscript technically sound, and do the data support the conclusions?

Reviewer #3: Yes

3. Has the statistical analysis been performed appropriately and rigorously? 

Reviewer #3: Yes

4. Have the authors made all data underlying the findings in their manuscript fully available?

Reviewer #3: Yes

5. Is the manuscript presented in an intelligible fashion and written in standard English?

Reviewer #3: Yes

6. Review Comments to the Author

Reviewer #3: Textual analysis under NLP/ML is quite hot in digital humanities area now. From my view, the data sources and research methodology of this research is innovative. This study focused on a very interesting topic about human–nature relationships revealed by AI. There is no doubt that the novel approaches and findings have potential for great contribution to digital humanities. However, there are a few suggestions for further refinement as follows:

(1)While the workflow diagram or flowchart was incorporated, the technical processes are still not clear enough. Its technical and process illustrations still need to be improved.

(2)In line 336,"(see Figure 1)"Perhaps needs to be modified to "(see Figure 2)".The same errors also occur in line 358 (see Figure 2) and Line 404 (see Figure 3).

In conclusion, I look forward to its eventual publication.

7. PLOS authors have the option to publish the peer review history of their article (what does this mean?). If published, this will include your full peer review and any attached files.

Reviewer #3: No
